# Soil Quality Assessment of Several Kinds of Typical Artificial Forestlands in the Inner Mongolia Basin of the Yellow River

Jiazheng Zhu [1,2,3,†], Zhenqi Yang [1,2,†], Fucang Qin [3,*], Jianying Guo [1,2], Tiegang Zhang [1,2] and Ping Miao [4]

1   Yinshanbeilu National Field Research Station of Steppe Eco-Hydrological System, China Institute of Water Resources and Hydropower Research, Beijing 100038, China
2   Institute of Water Resources for Pastoral Area, Ministry of Water Resources, Hohhot 010020, China
3   Desert Science and Engineering College, Inner Mongolia Agricultural University, Hohhot 010018, China
4   Ordos River and Lake Protection Center, Erdos 017200, China
*   Correspondence: qinfc@126.com; Tel.: +86-13704782159
†   Jiazheng Zhu and Zhenqi Yang are co-first authors.

**Abstract:** Located in the middle and upper parts of the Yellow River Basin, Qingshuihe County, Inner Mongolia, is a typical hilly and gully loess region and one of areas under the implementation of major ecological protection and restoration projects in the key ecological areas of Yellow River. Scientifically and accurately constructing a soil quality evaluation system for different types of artificial forest land and evaluating their soil quality are essential because they help optimize the structure of artificial forest land and improve the soil quality in the loess hilly area of Yellow River Basin. In this study, soil from four representative types of artificial forest land in the middle and upper reaches of the Yellow River Basin in Inner Mongolia was selected as the study object, with natural non-forest land as the control. Sixteen soil property indices in five classes, namely, soil texture, acid–base properties, moisture, pore, and nutrient, were screened using correlation analysis, minimum dataset (MDS), and principal component analysis methods, and an MDS of soil quality evaluation was constructed. Results showed that (1) the evaluation indices of artificial forest land soil quality based on the MDS included total potassium content, total phosphorus content, alkali-hydrolyzed nitrogen content, total nitrogen content, sand content, moisture content, and non-capillary porosity. (2) No significant differences were observed in the soil quality index among the MDS, total dataset, and significant dataset (SDS), all of which exhibited significant positive correlations. (3) The soil quality of the different types of forest land was sorted from high to low as follows: *mixed coniferous and broad-leaved* forests, *larch* forests, *mixed arbor and shrub* forests, *Armeniaca sibirica* forests, and natural non-forest land.

**Keywords:** Yellow River Basin; artificial forest land; minimum dataset; soil quality



## 1. Introduction

In recent years, intensified man-made destructive activities have led to the deterioration of the ecological environment on both sides of the middle and upper reaches of the Yellow River, the decline of forest resources, and the aggravation of regional water and soil loss, especially in the Inner Mongolia Basin of the Yellow River. In this context, the concept of artificial forests has been widely considered in research on improving the ecological environment of river basins. As a type of forest resource formed by artificial cultivation, artificial forests mainly function by improving the ecological environment, preventing soil erosion, preserving species resources, and promoting scientific experiments [1]. Today, artificial forests have become the main ecological restoration measure in the Inner Mongolia Basin of the Yellow River.

As an indispensable part of the forest ecosystem, soil is the root matrix and base of forests, and its complex structural composition has a direct impact on the survival and development of animals and plants and provides them with the material energy needed

for growth and development [2]. Therefore, scientifically and accurately evaluating the soil quality of artificial forest land is essential for assessing the sustainability of forest soil productivity and the stability of resistance. In current evaluations of forest soil quality, the physical and chemical properties and the biological characteristics of soil are often used as the main evaluation indices [3]. However, the traditional evaluation system based on a single factor cannot describe the comprehensive properties of soil systematically and comprehensively, so multiple factors must be selected to form a dataset for establishing a comprehensive soil quality evaluation system. Theoretically, a larger number of indexes can better reflect the overall soil quality. Chinese and foreign scholars have recently established evaluation index systems for several types of areas, such as farmland, woodland, and grassland, with more than 20 indexes classified into three categories, namely soil chemical indexes, physical indexes, and biological indexes [3,4]. However, such indexes are usually correlated, which easily leads to redundancy between index data; furthermore, too many measured indexes will consume considerable manpower and material resources [4]. Therefore, the evaluation indexes for soil quality should be selected more simply, economically, and rapidly.

With this in mind, Larson et al. [5] proposed the concept of the minimum dataset (MDS) in 1991, which included screening soil physical and chemical indexes based on principal component analysis (PCA). In this scheme, the MDS in soil quality evaluation is determined by combining correlation analysis and norm value calculations. Since then, extensive research has proven the suitability of the MDS in accurately evaluating soil quality. Brejda et al. [6,7] investigated plateaus in the central and southern parts of the United States and the Mississippi Loess Mountain and Palouse Prairie. Five–six representative indexes among twenty regional soil property indexes were screened out via factor analysis and discriminant analysis, allowing for the evaluation of regional soil quality, including total organic carbon (TOC), total nitrogen (TN), water-stable aggregate (WSA) content, potential mineralizable nitrogen (PMN), soil microbial biomass (MBC), and soil salinity. Zhou W T et al. [8] established an MDS system by using three methods—cluster analysis, principal component analysis, and correlation analysis—for soil quality evaluation and achieved high test precision. At present, red soil and black soil areas are evaluated for soil quality by using the MDS, but the opposite is true for arid and semi-arid areas.

The public welfare forest area in Qingshuihe County, Inner Mongolia, which is the study area of this research, is located at the junction of Mongolia, Shaanxi, and Shanxi. The loess hilly area in the middle part of the Yellow River Basin is innately deficient in natural water resources and has serious soil and water loss and a fragile ecological environment. As an important ecological security barrier in the arid and semi-arid areas of Northwest China, this area has been committed to existing forest resource protection; the physical and chemical properties of soil are taken as the key factors affecting the growth, development, and distribution of trees, and they are used as the key indexes to evaluate the service function of artificial forest ecosystems in the area and even the entire Yellow River Basin [9]. To date, only a few studies have evaluated the soil quality of large-scale artificial forest land in arid and semi-arid areas in the middle and upper reaches of the Yellow River. In this study, the forest soil in four representative types of artificial forests in the loess hilly area was selected as the study object, and natural nonforest land was chosen as the control. Then, the soil quality index system of artificial forests was investigated, and the soil quality was evaluated. A high final verification accuracy was achieved. This study can provide scientific guidance and a basis for optimizing the artificial forest land structure on the middle and upper reaches of the Yellow River and improving the soil quality, thereby rendering a reference for MDS-based index system construction in follow-up soil quality evaluation.

## 2. Materials and Methods

### 2.1. Overview of the Study Area

The Yellow River, which originates from the Bayankala Mountains in Qinghai Province, China, traverses nine provinces, including Qinghai Province, Gansu Province, and the Inner Mongolia Autonomous Region, and finally flows into the Bohai Sea. The Yellow River Basin has a vast territory, spanning three terrain steps from west to east, covering the Qinghai–Tibet Plateau, Inner Mongolia Plateau, Loess Plateau, and Huanghuaihai Plain. The north-to-south width can reach the Qinling Mountains and Yinshan Mountains, with an overall basin area of 795,000 square kilometers, and the natural geographical environment varies greatly [10]. The middle and upper reaches of the Yellow River occupy an area of 428,000 square kilometers, accounting for more than 53% of the total river basin area, with an altitude of 1000–2000 m, including large geological units such as the Hetao Plain, Ordos Plateau, Loess Plateau, and Fenwei Basin [10]. The overall hydrological and sediment conditions are complex, accompanied by a loose soil texture and fragile ecological environment as a whole. For the study area (Figure 1), Qingshuihe County is located at the southern end of the Inner Mongolia Basin on the middle and upper reaches of the Yellow River (111°21′–112°07′ E, 39°35′–40°11′ N), with an average elevation of 1400 m. Gullies and plains coexist in the county. The terrain is mainly hilly and mountainous, and the mountainous area accounts for more than 65% of the county area. Since the implementation of the second phase of the natural forest protection project, the forest preservation area has exceeded 100,000 hectares, and the forest coverage rate has exceeded 32%. The main tree species are *Larix gmelinii* (Rupr.), *Caragana korshinskii*, *Armeniaca sibirica*, *Populus alba*, and so on. Furthermore, Qingshuihe County is located in the mid-temperate zone of China, with a temperate continental monsoon climate, which is characterized by a hot rainy season. The annual precipitation is over 80% and concentrated during June–September. The mean annual precipitation is 413.8 mm, and the average number of rainy days is 75. This area has four distinct seasons, with an annual average temperature of 7.1 °C. In addition, the following eight soil types exist in the area: chestnut soil, chestnut cinnamon soil, gray cinnamon soil, alluvial soil, aeolian sandy soil, swamp soil, saline soil, and rocky soil [11].

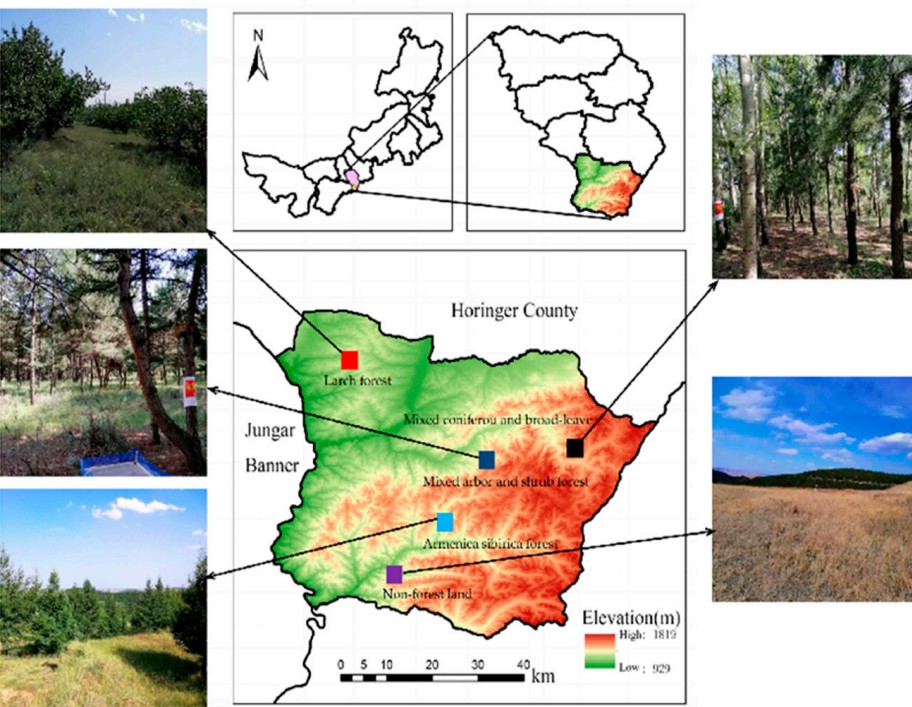

**Figure 1.** Study area location map.

*2.2. Index Determination*

2.2.1. Collection of Soil Samples

In July 2021, the undergrowth soil from four main types of artificial forests—*larch* forests, *Armeniaca sibirica* forests, *mixed arbor and shrub* forests (*Larix gmelinii + Caragana microphylla*), and coniferous and broad-leaved mixed forests (*Larix gmelinii + Abele*)—in the public welfare forest area of Qingshuihe County in the study area was selected as the study object. Then, natural non-forest land was selected as the control plot. Then, standard sampling plots were set in each forest land type. Factors such as topography, site conditions, and forest age were kept consistent. Furthermore, 20 sampling plots were established, with their sizes set to 15 m × 15 m. Finally, basic information such as altitude, tree height, DBH (root diameter), and canopy density was investigated and measured per plot. The altitude was measured using a GPS real-time measuring instrument. The canopy density was measured via the survey line method. In particular, with a representative section in the forest selected, the crown projection length of each tree along the 100-m survey line was measured, and the ratio of the total projection length of each crown on the survey line to the total length of the survey line was taken as the canopy density. Tree height and DBH were measured using a CriterionRD1000 dendrometer. The final measured basic information of the sampling plots is listed in Table 1.

**Table 1.** Basic information of the sample plots.

| Forest Land Type | Altitude (m) | Slope Aspect | Slope Position | Canopy Density (%) | Average Tree Height (m) | Average DBH (cm) | Main Tree Species |
|---|---|---|---|---|---|---|---|
| *Larch* forest | 920 | Northeast | Mid-slope | 61 | 8.77 | 14.60 | *Larch* |
| *Armenica sibirica* forest | 1420 | North | Mid-slope | 64 | 4.86 | 14.26 | *Armeniaca sibirica* |
| *Mixed coniferou and broad-leaved* forest | 1700 | Northeast | Mid-slope | 82 | 9.38 | 17.66 | *Larch* |
|  |  |  |  |  | 8.77 | 18.88 | *Abele* |
| *Mixed arbor and shrub* forest | 1650 | Northeast | Mid-slope | 57 | 7.21 | 19.20 | *Larch* |
|  |  |  |  |  | 1.93 | 3.19 | *Caragana microphylla* |
| Non-forest land | 1350 | Northeast | Mid-slope | — | — | — | — |

Five soil profiles (e.g., Figure 2) were dug using the five-point cross sampling method in each surveyed sample plot. The profile length and width was 1 m × 1 m, and the depth was 60 cm. On the premise of not damaging the soil structure, soil samples were collected using a cutting ring from 0–20, 20–40, and 40–60 cm soil layers, and the soil collection procedure was repeated three times for each layer [12]. The samples were recorded, numbered, weighed on site, sealed, and brought back to the laboratory.

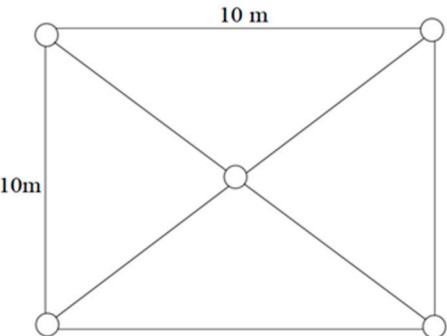

**Figure 2.** Five-point cross-sampling plot.

2.2.2. Determination of Index Values

(1) Soil moisture content was measured through the drying method [13]. The samples were dried in the laboratory at 105 °C for 12 h, their dry weight was weighed, and their moisture content was solved through the following formula:

$$W_1 = (W_s - W_c) \tag{1}$$

$$S = \frac{W_1 - W_0}{W_0} \times 100\% \tag{2}$$

where $W_1$ refers to soil it is either weight or mass (g); $W_s$ is the sample weight (g); $W_c$ is the weight of the hollow cutting ring (g); S is the soil moisture content (%); and $W_0$ is the dry soil weight (g).

(2) Soil bulk density and porosity were measured using the cutting ring method [14]. First, the weight and volume of the hollow cutting ring were recorded. Each ring cutter sample was soaked in water for 12 h, weighed, recorded, placed on dry sand for 12 h, and then weighed and recorded again. Second, the sample was placed in an oven to dry, and its dry weight was computed (105 °C, 12 h). The index calculation formula was as follows:

$$G = \frac{W_0}{V} \tag{3}$$

$$N1 = \frac{W_{12h} - W_0}{V} \times 100\% \tag{4}$$

$$N_2 = \frac{W_{12} - W_{12h}}{V} \times 100\% \tag{5}$$

where G is the soil bulk density (g/cm$^3$); V is the volume of the cutter ring (cm$^3$); $N_1$ is the capillary porosity of soil (%); $N_2$ is the non-capillary porosity of soil (%); N is the total porosity of soil (%); $W_{12}$ is the soil mass after soaking for 12 h (g); $W_0$ is the dry soil weight after drying (g); and $W_{12h}$ is the soil mass after being placed on dry sand for 12 h (g).

(3) Particle size distribution in soil: This variable was measured using the Malvin Mastersizer2000 laser particle size analyzer. The instrument mainly uses a verified laser diffraction system to measure the soil particle size. By measuring the diffraction light intensity distribution produced by the sample at different angles, the soil particle size distribution can be inversely calculated on the basis of the Michaelis scattering theory.

(4) Soil pH value was obtained through colorimetric determination using mixed indicator paper.

(5) Soil organic matter content was determined with the potassium dichromate-sulfuric acid solution oxidation method, and the specific determination process was as follows.

① A dry soil sample (0.5 g) sieved through a 0.25-mm sieve was placed in a test tube. $K_2Cr_2O_7$-$H_2SO_4$ solution was dropwise-added to the tube and evenly mixed, and a funnel was inserted at the tube mouth to ensure unblocked internal and external gas flows.

② A wire cage was inserted into the test tube containing the mixture of soil and solution, and the tube was placed in an oil bath pan at 190 °C. Then, the temperature was lowered to 170 °C, and timing was performed after the solution in the test tube started boiling. Notably, the solution was kept boiling but not intensely. After 5 min, the wire cage was removed and cooled for a while, and the oil droplets outside the test tube were wiped.

③ All the remaining liquid and soil residue in the boiled test tube were transferred to a triangular flask. The sticky substance, if any, on the inner wall of the test tube and small funnel was rinsed with distilled water, and washing liquor was poured into the triangular flask to ensure that the total volume of the solution was between 50 and 60 mL. Three droplets of o-phenanthroline indicator were dropwise-added to the triangular flask,

followed by continuous titration of the remaining $K_2Cr_2O_7$ with $FeSO_4$ standard solution until the titration end point. The calculation formula was as follows:

$$OM = \frac{c \times (V_0 - V) \times 0.003 \times 1.724 \times 1.1}{M} \times 1000 \qquad (6)$$

where OM refers to soil organic matter content (g/kg); $V_0$ is the volume of the $FeSO_4$ standard solution consumed via the blank experiment (mL); V is the volume of the $FeSO_4$ standard solution consumed via sample determination (mL); c is the concentration of the $FeSO_4$ standard solution (mol/L); and M is the dry weight of the soil samples (g).

(6) Total phosphorus and total potassium in soil: This variable was measured using the WD-J200 laser spectral element analyzer. The working principle of this instrument can be described as follows: The laser generated by the solid-state laser of the analyzer acts on the sample surface through the optical path system. When the laser energy is greater than the breakdown threshold energy of the sample, plasmas are formed on the sample surface. During the rapid back-motion to the low-energy state, the sample substances in these plasmas, which had been excited by laser energy, emit spectra with information on the types and content of the sampled elements. These emission spectral signals are collected by an intelligent signal collection system, transmitted to the spectrometer for splitting, and detected using a CCD detector to obtain the information on the required element content.

(7) Soil total nitrogen content was determined through the Kjeldahl distillation method, and the specific determination process was as follows.

① Dry soil (1 g) sieved with a 0.25-m sieve was placed in the bottom of a digestive tube. Then, 1.8 g of accelerator, 2 mL of water, and 5 mL $H_2O_4$ (concentrated) were added and mixed evenly. The mixture was placed on a temperature-controlled digester and heated on low fire for 15 min. When the reaction tended to be moderate, the sample was heated to 375 °C. After the digestion liquid and soil particles turned gray and slightly green, digestion was continued for 1 h. Afterward, the digestion liquid was cooled to room temperature for subsequent use.

② After the digestion liquid was cooled, the cleaned digestive tube was added with 60 mL of water, shaken evenly, and placed on a nitrogen determinator. Then, 25 mL of boric acid–indicator mixed solution was added to the triangular flask, the orifice of the condenser adapting pipe was placed below the indicator level, 35 mL of the original NaOH solution was slowly added to the digestive tube, and distillation was performed for 5 min. The distillate was titrated with 0.01 mol concentration of the HCL standard solution. When it turned from blue-green to red-purple, the volume of the acid standard solution used was recorded immediately. The calculation formula is as follows:

$$\Omega = C \, (V - V_0) \times 14 \times M \qquad (7)$$

where $\Omega$ is the total nitrogen content (g/kg); C is the concentration of the standard hydrochloric acid solution (mol/L); V is the volume of the standard hydrochloric acid solution consumed by the sample (mL); $V_0$ is the volume of the standard hydrochloric acid solution consumed by the blank experiment (mL); 14 is the molar mass of oxygen; and M is the dry measure of the soil sample (g).

(8) The rapidly available phosphorous content in soil was determined through $NaHCO_3$ digestion–Mo-Sb colorimetry, and the specific determination process was as follows.

① Dry soil (2.5 g) sieved with a 2-m sieve was weighed and placed in a plastic bottle, and 50 mL of $NaHCO_3$ extracting agent was added.

② Next, 0, 0.5, 1, 2, 3, 4, and 5 mL of the standard solution were absorbed into a volumetric flask, and 10 mL of the $NaHCO_3$ extracting agent and 5 mL of the Mo-Sb color development agent were added and mixed evenly. After all of the $CO_2$ was discharged, water was added to a constant volume, the room temperature was kept below 20 °C, and the sample was allowed to stand for 30 min. At the wavelength of 880 nm, colorimetric

determination was performed using a 1-cm cuvette in optical length after zero setting through the zero point of the standard solution. Then, the standard curve was drawn.

③ The sample solution (10 mL) was absorbed into a 50-mL volumetric or conical flask, and 5 mL of the Mo-Sb color development agent was slowly added and mixed evenly. After all of the $CO_2$ was discharged, 10 mL of water was added, and the sample was allowed to stand for 30 min. At the wavelength of 880 nm, colorimetric determination was performed using a 1-cm cuvette in optical length after zero setting through the zero point of the standard solution. The calculation formula was as follows:

$$\omega = \frac{(P - P_0) \times V \times D}{M \times 1000} \times 1000 \tag{8}$$

where $\omega$ is the rapidly available phosphorous content in soil (mg/kg); P is the phosphorous content in the chromogenic solution (mg/L); $P_0$ is the phosphorous concentration in the blank sample (mg/L); V is the volume of the chromogenic solution (mL); D is the ratio of the volume of the sample extracting agent to fractional volume; and M is the mass of the soil sample (g).

(9) The rapidly available potassium content in soil was determined through the neutral ammonium acetate leaching method, and the specific determination process was as follows.

① Five grams of dry soil that had passed through a 1-m sieve was placed in a plastic bottle, and 50 mL of $CH_3COONH_4$ solution was added. The bottle stopper was plugged into the bottle mouth and tightened. The sample was oscillated at a speed of 180 r/min for 30 min at 20–25 °C, dried, and filtered. Then, the filtrate was directly determined using a flame photometer.

② Next, 0, 3, 6, 9, 12, and 15 mL of the standard K solution were absorbed into a 50 mL volumetric flask, and the volume was kept constant by using a $CH_3COONH_4$ solution. The zero point of the instrument was regulated with the K solution at a concentration of 0, and its value was determined with a flame photometer. The calculation formula was

$$K = (CK \times V)/M \tag{9}$$

where K is the rapidly available potassium content in soil ($mg \cdot kg^{-1}$); CK is the concentration of K in the undetermined solution (μg/mL); V is the volume of the extracting agent (mL); and M is the sample mass (g).

(10) Soil alkali-hydrolyzed nitrogen content was determined through the alkaline hydrolysis diffusion method, and the specific operation method was as follows.

① Dry soil (2 g) that had passed through a 2-m sieve was weighed and spread on the outer chamber of the diffusion dish, and 1 g of Zn-FeSO$_4$ powder was weighed and spread evenly on the soil sample.

② Three milliliters of the boric acid solution indicator was added to the inner chamber of the diffusion dish, and alkaline glycerin was spread evenly above the edge of the outer chamber of the diffusion dish, which was covered with ground glass and bonded to reveal a gap. Then, 10 mL of 1.8 mol/L$^{-1}$ NaOH solution was dropwise-added to the outer chamber of the diffusion dish. The ground glass was immediately placed tightly, the diffusion dish was rotated horizontally so that the solution and soil samples were fully mixed, and it was fixed with rubber bands.

③ The diffusion dish was placed in a thermostat at 40 °C for 24 h. After taking it out, 0.01 mol/L$^{-1}$ standard saline solution was dropwise-added to the boric acid indicator in the inner chamber of the diffusion dish with a micro burette until the reagent turned purple. In the experiment, another diffusion dish was used for the synchronous blank experiment, and the final value was recorded.

### 2.3. Soil Quality Evaluation

2.3.1. Establishment of the Minimum Dataset

The data of 16 indexes measured in Section 2.2.2 were divided into the following five categories: soil texture, pH, moisture, pores, and nutrients. On this basis, a primary index

dataset [total dataset (TDS)] was constructed. After comparing the coefficient of variation, the indexes with low differences in the TDS were eliminated, and the remaining indexes were formed as a significant dataset (SDS). The factor loading of principal components was extracted using SPSS 19.0 (http://www.spss.com.cn, accessed on 23 April 2023), and a PCA was adopted to calculate the norm value and reflect the relative importance of each index in the SDS in multidimensional space. The formula for calculating the Norm value is given using

$$N_{ik} = \sqrt{\sum_{i=1}^{k}(U_{ik}^2 \beta_k)} \tag{10}$$

where $N_{ik}$ is the comprehensive load of the i-th variable in the first k principal components with an eigenvalue; $U_{ik}$ is the load of the i-th variable in the k-th principal component; and $\beta_k$ is the eigenvalue of the k-th principal component.

The indices were grouped by following the rule that the absolute value of the factor loading of each principal component was greater than 0.5. If the factor loading of an index in multiple principal components was greater than 0.5, this index could be included using the correlation analysis method into a group showing minor correlations with other indices. After grouping, the indices whose Norm values fell within the range of 10% of the maximum value were subjected to a correlation analysis. When the correlation coefficient between indices was greater than 0.5, the two indices were considered to be highly corelated, and the indexes with high Norm values were selected in the MDS. On the contrary, when all correlation coefficients were smaller than 0.5, all indices were included in the MDS [15,16].

### 2.3.2. Calculation of the Soil Quality Index

Given the difference in the dimensional data of each to-be-evaluated index, the problem of their unified standardization should be solved to realize the data unification of all indices and make them comparable [17]. Therefore, the membership function of each index should be created, and the membership function value of each index should be calculated. In this study, two membership functions (Formulas (2) and (3)), namely ascending-type and descending-type membership functions, were selected.

Ascending membership function:

$$F(x) = \begin{cases} 0.1 & x < x_1 \\ 0.1 + 0.9\,(x - x_1)/(x_2 - x_1) & x_1 < x < x_2 \\ 1.0 & x > x_2 \end{cases} \tag{11}$$

Descending membership function:

$$F(x) = \begin{cases} 0.1 & x > x_2 \\ 0.1 + 0.9\,(x_2 - x)/(x_2 - x_1) & x_1 < x < x_2 \\ 1.0 & x < x_1 \end{cases} \tag{12}$$

In the equations, x is the actual value of the evaluation index, $x_1$ is the minimum index value, $x_2$ is the maximum index value, and $x_1$ and $x_2$ are the values of the turning point on the functional curve.

The weighted coefficient of each index was calculated using the ratio of the common factor variance of each index to the sum of the common factor variance of all indices calculated using the PCA method [18]. The calculation formula of soil quality index (SQI) is as follows:

$$SQI = \sum_{i=1}^{n} N_i X_i \tag{13}$$

where $N_i$ is the weight coefficient of the i-th index; $X_i$ is the membership value of the i-th index; and n is the number of indices.

## 3. Results

### 3.1. Analysis of the Soil Property Differences between Different Types of Artificial Forest Land in Inner Mongolia Basin of the Yellow River

The soil properties displayed evident differences with the increase in the soil depth. Therefore, the arithmetic average of three layers of soil data in each plot was calculated to objectively compare the soil property differences under forest land [19], as shown in Table 2.

**Table 2.** Eigenvalues of soil indices in each type of forest land.

| Type | Index | Pure Forest | | Mixed Forest | | Non-Forest Land | Coefficient of Variation (%) |
| --- | --- | --- | --- | --- | --- | --- | --- |
| | | *Larch* | *Armeniaca sibirica Forest* | *Mixed Arbor and Shrub Forest* | *Mixed Coniferous and Broad-Leaved Forest* | | |
| Texture type | Clay content (%) | 12.88 ± 1.36 | 4.57 ± 0.99 | 4.06 ± 2.39 | 9.98 ± 3.68 | 2.74 ± 0.85 | 57.87 |
| | Sand content (%) | 53.35 ± 7.88 | 69.33 ± 5.21 | 68.42 ± 3.96 | 56.49 ± 9.93 | 70.79 ± 3.46 | 11.38 |
| | Silt content (%) | 34.11 ± 3.29 | 28.42 ± 1.17 | 28.51 ± 4.78 | 34.86 ± 2.77 | 28.66 ± 6.19 | 9.45 |
| | Soil bulk density | 1.36 ± 0.18 | 1.43 ± 0.33 | 1.24 ± 0.26 | 1.19 ± 0.25 | 1.49 ± 0.65 | 9.88 |
| Acid-base property type | pH | 8.40 ± 0.03 | 8.69 ± 0.10 | 8.29 ± 0.34 | 8.26 ± 0.09 | 7.6 ± 0.82 | 4.35 |
| Moisture type | Moisture content (%) | 12.79 ± 2.33 | 7.04 ± 1.18 | 10.19 ± 5.35 | 15.64 ± 6.20 | 5.52 ± 4.42 | 43.36 |
| Pore type | Total porosity (%) | 63.55 ± 7.65 | 51.88 ± 8.96 | 43.98 ± 3.45 | 68.17 ± 1.99 | 26.42 ± 10.85 | 31.82 |
| | Capillary porosity (%) | 56.76 ± 5.28 | 46.57 ± 4.96 | 36.94 ± 1.79 | 59.71 ± 0.83 | 23.58 ± 6.92 | 33.22 |
| | Non-capillary porosity (%) | 6.78 ± 1.26 | 5.31 ± 3.33 | 7.04 ± 0.85 | 8.45 ± 0.17 | 2.84 ± 0.59 | 30.96 |
| Nutrient type | Organic matter content (g·kg$^{-1}$) | 18.54 ± 3.28 | 11.69 ± 2.10 | 16.08 ± 2.25 | 27.74 ± 2.16 | 5.55 ± 0.98 | 47.29 |
| | Total phosphorous content (g·kg$^{-1}$) | 0.84 ± 0.03 | 0.45 ± 0.16 | 0.62 ± 0.08 | 0.90 ± 0.36 | 0.41 ± 0.02 | 31.02 |
| | Total potassium content (g·kg$^{-1}$) | 17.15 ± 3.66 | 10.59 ± 2.76 | 14.01 ± 0.89 | 20.99 ± 2.33 | 5.19 ± 2.24 | 41.38 |
| | Total nitrogen content (g·kg$^{-1}$) | 0.87 ± 0.01 | 0.52 ± 0.06 | 1.41 ± 0.03 | 1.55 ± 0.19 | 0.46 ± 0.04 | 45.83 |
| | Rapidly available phosphorous content (mg·kg$^{-1}$) | 13.88 ± 1.98 | 10.01 ± 1.58 | 19.62 ± 1.28 | 22.70 ± 3.68 | 3.77 ± 0.06 | 48.24 |
| | Rapidly available potassium content (mg·kg$^{-1}$) | 113.62 ± 9.05 | 130.85 ± 8.76 | 149.96 ± 3.28 | 133.17 ± 3.76 | 50.25 ± 4.99 | 33.92 |
| | Alkali-hydrolyzed nitrogen content (mg·kg$^{-1}$) | 98.66 ± 9.99 | 72.38 ± 6.49 | 60.15 ± 6.66 | 118.95 ± 9.32 | 30.33 ± 1.29 | 40.32 |

Note: The data in the above data are arithmetic mean ± standard error.

As shown in Table 2, the average soil pH is 8.25, indicating that the soil in the study area is alkaline. Except for the rapidly available potassium content, the values of the nutrient indexes of *mixed coniferous and broad-leaved* forests were higher than those of other types of forest land, and the available potassium content was the highest in the mixed arbor–shrub forest (149.96 mg kg$^{-1}$). The ability of forest soil to provide nutrients for vegetation was higher than that of non-forest land, and the overall fertility status of soil under *mixed coniferous and broad-leaved* forests was better than that under the other types of forest land. The specific soil physical property indices of each type of forest land showed that among all the types of forest land, the total porosity; capillary porosity; non-capillary porosity; silt content; and moisture content of the *mixed coniferous and broad-leaved* forest land were the highest, and the bulk density was the lowest; the percentages were 68.17%, 59.71%, 8.45%, 34.86%, 15.64%, and 1.19%, respectively. This result indicates that the soil structure of the *mixed coniferous and broad-leaved* forest land was looser and more ventilated than that of the other types of forest land. The highest clay content in *larch* forests was 12.88%, and the highest sand content in non-forest land was 70.79%. The abovementioned indices explained the soil quality of each type of forest land from different angles, but the

evaluation results of each index had no unified direction, making it impossible to judge which forest land type had better soil quality.

With regard to the sensitivity classification of comprehensive evaluation indices for soil quality proposed by Mingxiang Xu et al. [20] a coefficient of variation ≥100% indicates high sensitivity, 100–40% indicates moderate sensitivity, 40–10% indicates low sensitivity, and ≤10% indicates insensitivity. As calculated from the data in Table 2, the clay content (57.87%), moisture content (43.36%), organic matter content (47.29%), total potassium content (41.38%), total nitrogen content (45.83%), rapidly available phosphorus content (48.24%), and alkali-hydrolyzed nitrogen content (40.32%) were moderate-sensitivity indices. Among them, the soil clay content reached the maximum coefficient of variation and the highest data discreteness degree. Soil sand content (11.38%), total porosity (31.82%), capillary porosity (33.22%), non-capillary porosity (30.96%), total phosphorus content (31.02%), and rapidly available potassium content (33.92%) were low-sensitivity indices. The coefficients of variation for the soil silt content (9.45%), pH (4.35%), and bulk density (9.88%) were below 10%, revealing that these indices were concentrated and insensitive with a low discreteness degree. Hence, the three indices were unsuitable for the evaluation of soil quality.

### 3.2. Establishment of MDS and Determination of Index Weights

The three low-sensitivity indices, namely soil silt content, pH, and bulk density, with sensitivities < 10%, were excluded, and the remaining 13 indices constituted an SDS for PCA. The PCA results (Table 3) showed that the accumulative variance contribution rate of the first four principal components was 78.420%, revealing that they could completely explain the soil property information under artificial forest land in the study area.

**Table 3.** Principal component factor loading and Norm value of soil indices.

| Type | Index | Group | Principal Component | | | | Norm |
|---|---|---|---|---|---|---|---|
| | | | **1** | **2** | **3** | **4** | |
| | Total potassium | 1 | 0.836 | 0.15 | −0.256 | −0.164 | 1.979 |
| | Total phosphorous | 1 | 0.78 | −0.068 | 0.426 | 0.124 | 1.882 |
| | Alkali-hydrolyzed nitrogen | 1 | 0.759 | −0.194 | −0.054 | 0.454 | 1.847 |
| Nutrient type | Rapidly available phosphorous | 1 | 0.733 | 0.008 | −0.228 | −0.44 | 1.781 |
| | Organic matter | 1 | 0.711 | −0.357 | −0.037 | −0.388 | 1.780 |
| | Rapidly available potassium | 1 | 0.709 | 0.577 | −0.16 | 0.009 | 1.865 |
| | Total nitrogen | 4 | 0.499 | 0.418 | 0.496 | −0.509 | 1.542 |
| Texture type | Clay | 1 | 0.534 | −0.49 | 0.146 | −0.279 | 1.479 |
| | Sand | 2 | −0.455 | 0.747 | 0.054 | −0.03 | 1.541 |
| Moisture type | Moisture content | 3 | 0.42 | −0.067 | −0.685 | 0.346 | 1.327 |
| | Total porosity | 1 | 0.775 | 0.338 | −0.178 | 0.138 | 1.881 |
| Pore type | Non-capillary porosity | 2 | 0.477 | 0.611 | 0.347 | 0.281 | 1.525 |
| | Capillary porosity | 2 | 0.419 | −0.521 | 0.452 | 0.402 | 1.425 |
| | Eigenvalue | - | 5.352 | 2.262 | 1.424 | 1.157 | - |
| | Variance contribution rate (%) | - | 41.172 | 17.379 | 10.952 | 8.899 | - |
| | Accumulative contribution rate (%) | - | 41.172 | 58.569 | 69.521 | 78.420 | - |

As shown in Table 3, the soil indices could be divided into four groups in accordance with the absolute values of the load of each principal component factor. Soil total potassium content, total phosphorus content, alkali-hydrolyzed nitrogen content, rapidly available phosphorus content, organic matter content, rapidly available potassium content, clay content, and total porosity were included in the first group, which mainly reflected the soil fertility status. Soil sand content, capillary porosity, and non-capillary porosity constituted the second group, which reflected the soil texture. The third and fourth

groups each contained only one index, namely the moisture content and total nitrogen content, respectively.

According to the establishment principle of the MDS [16,21], the Norm values of organic matter content and clay content in the first group of indices were not within the range of 10% of the highest Norm value in the first group, so they were not included in the MDS. The correlation analysis of the remaining indices showed that in the first group, soil total potassium content was highly correlated with rapidly available phosphorus content and alkali-hydrolyzed nitrogen content (correlation coefficient > 0.5), and the Norm value of rapidly available phosphorus content was the lowest. The correlation between soil rapidly available potassium content and total porosity was high, but the Norm value of the former was lower than that of the latter. The correlation between soil total phosphorus content and total porosity was high, but the Norm value of the latter was lower than that of the former. Therefore, rapidly available phosphorus content, rapidly available potassium content, and total porosity were not included in the MDS. Similarly, sand content and non-capillary porosity in the second group were included in the final MDS. The third and fourth groups each had only one index, both of which were included in the final MDS. Seven MDS indices, namely total potassium content, total phosphorus content, alkali-hydrolyzed nitrogen content, total nitrogen content, sand content, moisture content, and non-capillary porosity, were finally selected.

### 3.3. Soil Quality Assessment of Different Types of Artificial Forest Land in Inner Mongolia Basin of the Yellow River

The index weight values (e.g., Figures 3–5) in each dataset were calculated in accordance with the proportion of the total variance of the common factors of the indices obtained through the PCA method.

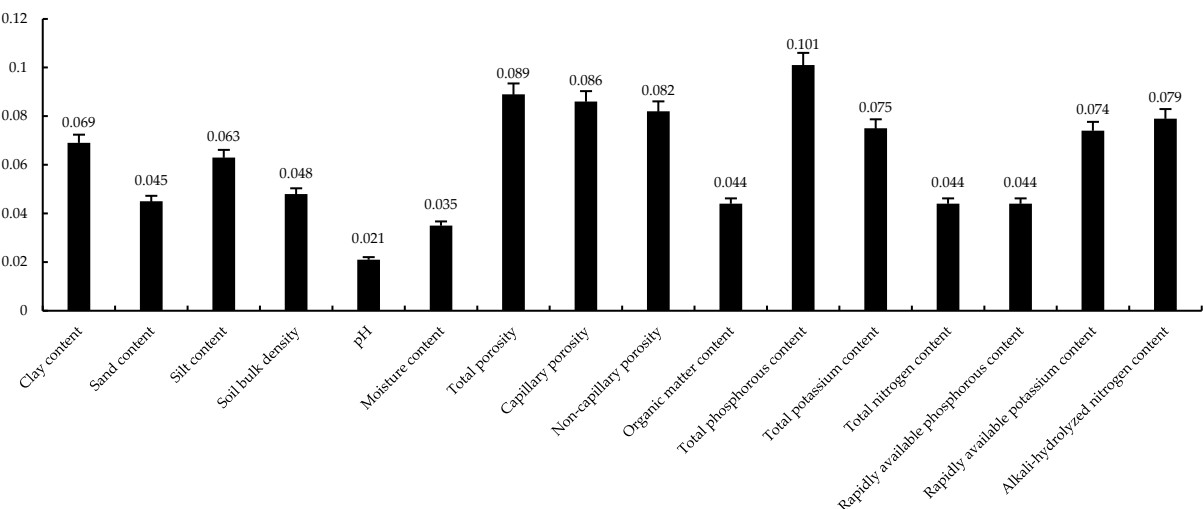

**Figure 3.** TDS index weight values.

In accordance with the positive and negative factor loading of each index on the principal component, the membership function type was selected, and the membership value was calculated. The soil physical and chemical comprehensive index (SQI) was calculated with the weighted summation method, and the quality index of TDS, SDS, and MDS was obtained [22]. To verify the rationality of the MDS, the quality index of all sample plots in the MDS, TDS, and SDS was subjected to regression analysis, and the results are displayed in Figures 6 and 7.

The regression analysis results showed that the MDS and TDS had a linear relation of $y = 0.8579x + 0.1232$, where $R^2$ was 0.8083. The MDS and SDS presented a linear relation of $y = 0.964x + 0.0606$, where $R^2$ was 0.8718, which was greater than 0.8 and indicated a high degree of fitting. The MDS was significantly positively correlated with the TDS and SDS,

revealing that the MDS established in this study could completely represent the evaluation information of TDS and SDS indices for soil quality in different types of artificial forest land in the study area. Therefore, the established MDS index system could effectively reflect the soil quality under different types of artificial forest land in the study area.

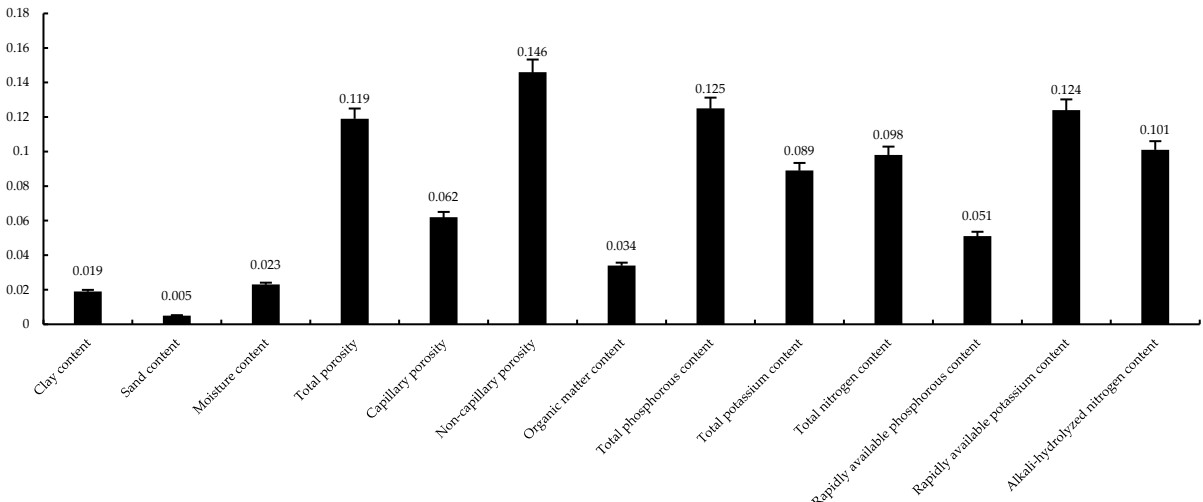

**Figure 4.** SDS index weight values.

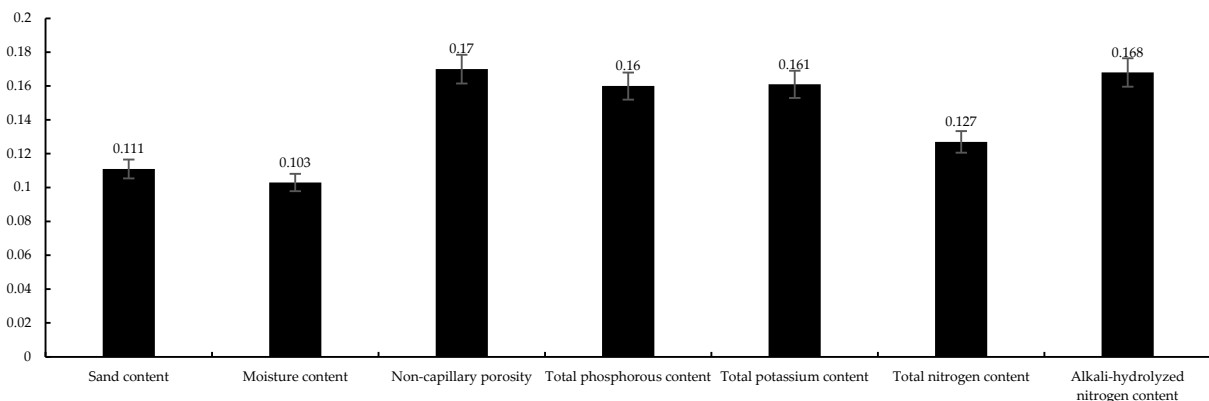

**Figure 5.** MDS index weight values.

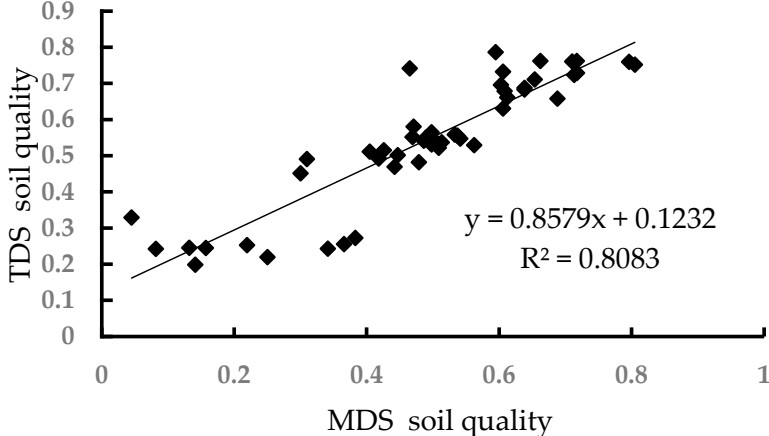

**Figure 6.** Correlation between soil quality indices in the MDS and TDS.

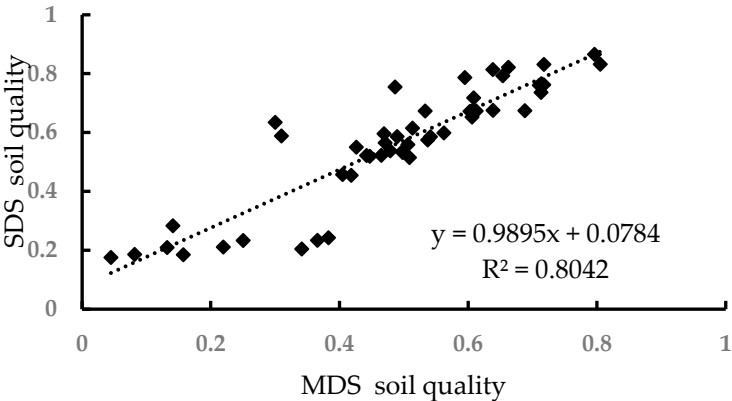

**Figure 7.** Correlation between soil quality indices in the MDS and SDS.

Figure 8 shows a data map of eight MDS indices of soil in each type of artificial forest land in the Inner Mongolia Basin of the Yellow River. The area formed by connecting the eight data points was the comprehensive SQI of each type of artificial forest land. Figure 8 shows that the soil total phosphorus content, total potassium content, alkali-hydrolyzed nitrogen content, moisture content, and non-capillary porosity of the coniferous and broad-leaved mixed forest had the highest values and constituted the largest area on the radar map. Non-forest land had the minimum values of soil total potassium content, total nitrogen content, alkali-hydrolyzed nitrogen content, moisture content, and non-capillary porosity. In accordance with the geometric meaning of the radar map [23] (the farther the coordinate points are from the origin, the better the state of the index is; the larger the area surrounded by each coordinate point is, the better the overall state of the evaluation object is), soil quality was sorted from good to poor as follows: the *mixed coniferous and broad-leaved* forest > *larch* forest > *mixed arbor and shrub* forest, *Armeniaca sibirica* forest, and natural non-forest land.

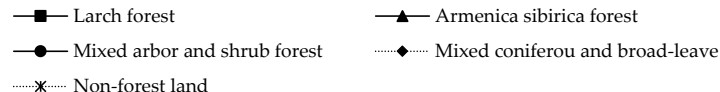

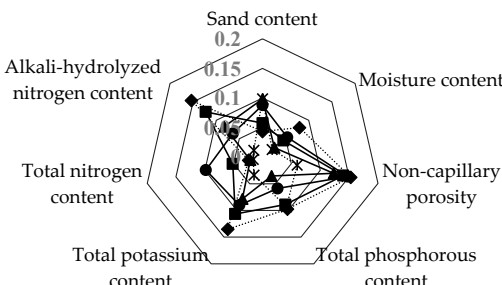

**Figure 8.** Soil quality index of different types of forest land.

## 4. Discussion

The overall soil quality, an important matrix of forest ecosystems, plays an important role in the growth and distribution of trees. The international definition of soil quality as used presently was proposed by Parkin and Doran [24] in 1994, which is the "ability of soil to maintain environmental quality, keep biological productivity, and promote the health of animals and plants in ecosystems." Studies in China and abroad have shown that soil quality is influenced by many factors, such as soil properties, climate environment, and

vegetation growth, making it particularly important to construct an evaluation system that can objectively and accurately reflect soil quality [25–27]. Incidentally, a unified soil quality evaluation system is currently lacking because of the different research areas and focal points. Studies have also shown that the primary loess area in the loess hilly region of the Yellow River in Inner Mongolia can reach 381,000 square kilometers, and the overall soil texture is soft and rich in mineral nutrients, among which the physical and chemical properties of soil play a major role in the growth and distribution of vegetation and the maintenance of soil biological activity [28]. On the basis of the basic physical and chemical properties of soil, the MDS used to evaluate the soil quality of different types of artificial forests in the basin was determined by combining the results of principal component analysis and correlation analysis.

The results revealed significant differences in the soil physical and chemical indexes of different land-use types in the Inner Mongolia Basin of the Yellow River. From the weight values of the MDS indexes, soil noncapillary porosity has the highest weight among all indexes; it is the first limiting factor of soil physical properties and directly impacts the overall quality of soil. Soil alkali-hydrolysable nitrogen has the largest weight among the nutrient classification indexes, with its content closely related to the soil nutrient environment; it can be used as the first factor to indicate forest soil fertility in the middle and upper reaches of the Yellow River in the future [29]. The comprehensive soil quality index (SQI) of each type of artificial forest land was analyzed on the basis of the MDS. Then, the soil quality in each type of artificial forest land was sorted from "good" to "bad" as follows: the *mixed coniferous and broad-leaved* forest, *larch* forest, *mixed arbor and shrub* forest, *Armeniaca sibirica* forest, and natural non-forest land. The overall performance of the mixed forest is better than that of the pure forest in improving soil quality, which coincides with the research results of Koupar et al. [30] on the soil quality of three forest types in northern Iran, namely the fast-growing pure forest of *Populus deltoides*, the pure Alnus forest, and the *Populus deltoides*–Alnus mixed forest with different proportions (70P:30A, 50P:50A, and 30P:70A). This research was undertaken over 13 years in the study area, suggesting convincing conclusions. A reason may be that the mixed forest has a multilayer canopy with a complex stand structure, which can form a more stable habitat and microclimate than the pure forest while providing favorable conditions for soil microorganism activities. Regarding forest structure optimization, Guo et al. [31] studied the interaction between the productivity of two *larch* species (*Larix kaempferi* and *L. olgensis*) and the corresponding undergrowth soil and found that starch and unstructured carbohydrates in mixed forest soil can increase significantly without fertilization, and mixed planting and fertilization can effectively improve the productivity of *larch*. This finding is consistent with the results of the present study. The planting proportion of *larch* mixed forests can be appropriately increased on the basis of local conditions in the future.

In the currently established MDS-based soil quality evaluation index systems in China and abroad, most indexes are selected from soil physical and chemical indexes [32]. Studies have shown that soil organisms directly participate in the transformation process of soil substances [33]. In this research, the study area belongs to the key control area of soil erosion in the Yellow River Basin in the north-central part of China. In a follow-up study, soil biological indexes, such as microbial number and soil microbial biomass, and the indexes affecting soil erosion performance, such as erosion resistance, erosion modulus, runoff, and erodibility, can be included in the primary index evaluation system to enhance the objectivity and universality of the established MDS. In addition, a positive correlation exists between soil nutrient content and forest age, as evidenced by a large number of studies [34]. The evaluation results of the current study are based on the consistency of the tree species' forest ages and site conditions. Given the limitations of regional climate and soil nutrient conditions, different tree species vary in their growth and degradation periods, and their effects on soil quality also differ. In future studies on forest soil quality in the middle and upper reaches of the Yellow River Basin, multiple sets of data samples

can be selected to explore the relationships between the forest age of different tree species and growth period and soil quality.

## 5. Conclusions

(1) On the basis of the MDS, the evaluation indices of soil quality under different types of artificial forest land in the Inner Mongolia Basin of the Yellow River included total potassium content, total phosphorus content, alkali-hydrolyzed nitrogen content, total nitrogen content, sand content, moisture content, and non-capillary porosity.

(2) No significant difference in soil quality index was observed among the MDS, TDS, and SDS, which presented significant positive correlations ($R^2 > 0.8$) in all cases, indicating that the established MDS could accurately reflect the soil quality information in the loess hilly region.

(3) The soil quality indices of different types of artificial forest land showed that the overall soil quality of the Yellow River Basin in Inner Mongolia belonged to the middle level. The soil quality of *mixed coniferous and broad-leaved* forests with complex canopies and developed underground root systems was the best. Therefore, in future artificial forest construction, the proportion of *mixed coniferous and broad-leaved* forests with poplar and *larch* as the main tree species can be appropriately increased based on the principle of planting trees under local conditions.

**Author Contributions:** J.Z. and Z.Y.; methodology, J.Z.; software, F.Q.; validation, J.Z.; formal analysis, J.Z.; investigation, J.Z.; resources, J.Z.; data curation, J.Z.; writing—original draft preparation, J.Z. and Z.Y.; writing—review and editing, J.G. and T.Z.; visualization, F.Q.; supervision, J.G. and P.M.; project administration, F.Q. and Z.Y.; funding acquisition. All authors have read and agreed to the published version of the manuscript.

**Funding:** This study was supported by the Major Science and Technology Projects of Inner Mongolia Autonomous Region (2020ZD0009), the "Science for a Better Development of Inner Mongolia" Program (KJXM-EEDS-2020005) of the Bureau of Science and Technology of the Inner Mongolia Autonomous Region, and the Major Science and Technology Projects of Inner Mongolia Autonomous Region (2021ZD0008).

**Data Availability Statement:** Not applicable.

**Conflicts of Interest:** The authors declare no conflict of interest.

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
