# Peer review of "Soil Quality Assessment of Several Kinds of Typical Artificial Forestlands in the Inner Mongolia Basin of the Yellow River"

_land, doi:10.3390/land12051024_

Round 1

Reviewer 1 Report

the comments for the manuscript reviewed are attached herewith

Author Response

The content of the manuscript shall be revised according to the review comments. Attached is the revised manuscript

Reviewer 2 Report

The authors did not follow MDPI citation format

The cited references are not serially arranged

In page 3 line 139, it should be written as (Figure 2 and 3)

In Equation 2, 3, 4 and 5, Kindly rewrite it using Equation.

In page 7, line 231, “5 mL of mLH2O(concentrated) were …..” should be re-written.

In page 7, line 242, “mol/LHCL” should be written as “mol/HCl”.

In page 7, line 246, ………”(g.kg-1)”, kindly maintain same format as previous. You should maintain either g.kg-1 or g/kg in the whole manuscript.

The conclusion is too lengthy. I suggest the authors to conclusion based on their result findings.

Author Response

1.Changed manuscript to MDPI reference format

2.The sequence of references has been revised

3.Figures 2 and 3 have been deleted

4.The equation has been rewritten

5.Formatting errors and conclusions have been corrected

Reviewer 3 Report

In general, the manuscript brings valuable information about soil quality, however, the authors need to improve the formatting (quality of images/figures) and better discuss the results.

Below are some specific comments:

Line 139: Check to space.

Figures 2, 3, and 4: are perhaps unnecessary in the manuscript, especially figure 3 where it is not possible to read the information (I think it’s written in Chinese or Cantonese).

Correct table formatting 1.

Correct formatting of formulas.

Correct the spacing of numbers in table 2.

Was a data normality test performed?

How was the choice of the number of principal components defined? Authors need to provide more details on the material and methods.

Authors generally need to check the formatting of figures and the numbers in them.

In figures 8 and 9, what is the significance of the correlation and what is the coefficient of determination? Please add the equation used in the adjustment.

Figure 10 it is not possible to identify the values of the indexes, the authors need to remake the figure.

Author Response

1.Figures 2 and 3 have been removed

2.The chart format and chart have been modified

3.Data normality test has been performed

4.Provides more detailed information on the number of principal components selected

5.Figures 8, 9, and 10 have been changed to Figures 6, 7, and 8, all modified

Reviewer 4 Report

I read through the manuscript called „Soil Quality Assessment of Several Kinds of Typical Artificial  Forest Land in the Inner Mongolia Basin of Yellow River“ by Jiazheng Zhu et al. I appreciate the presented work very much because soil quality in the forests is an important factor of their growth. Congratulations to a valuable manusript.

I have only some minor comments to the manuscript.

I miss any information about the soil itself, description of a soil profile, soil classification. Is there any difference in the soil types under different types of forests? Is the geology different? When talking about the soils and soil quality, appropriate soil names are needed.

Otherwise I think the experiment is well set up, the selection of study sites is explained. The results are clearly presented, which provides a good interpretation basis. 

The discussion and the conclusion parts of the manuscript are well worked out. The authors state themselves that good functioning of the models is based on good calibration so good measurements of field data.

Please enlarge the figure 10, it is not readable in this size.

I recommend the manuscript for publication.

Author Response

Thank the reviewer's valuable comments! More detailed soil information is provided in the article.

Round 2

Reviewer 1 Report

The author have done the necessary changes and can be accepted for publication

Author Response

Further adjustments have been made to the language.Thanks to reviewers!

Reviewer 3 Report

The authors made the requested corrections, however, there are some writing errors (like HCL on line 254), and for that reason, I recommend the publication of the manuscript after minor revisions. I suggest that the authors review again the writing of some sentences.

Author Response

The writing errors of some sentences have been corrected.Thank the reviewers for their valuable comments
